# The Effects of Epicuticular Wax on Anthracnose Resistance of *Sorghum bicolor*

**DOI:** 10.3390/ijms24043070

**Published:** 2023-02-04

**Authors:** Wangdan Xiong, Longxin Liao, Yu Ni, Hanchi Gao, Jianfeng Yang, Yanjun Guo

**Affiliations:** 1Qingdao Key Laboratory of Specialty Plant Germplasm Innovation and Utilization in Saline Soils of Coastal Beach, Qingdao Agricultural University, Qingdao 266109, China; 2Key Laboratory of National Forestry and Grassland Administration on Grassland Resources and Ecology in the Yellow River Delta, Qingdao Agricultural University, Qingdao 266109, China; 3College of Grassland Science, Qingdao Agricultural University, Qingdao 266109, China; 4College of Agronomy and Biotechnology, Southwest University, Chongqing 400715, China; 5College of Agronomy, Qingdao Agricultural University, Qingdao 266109, China

**Keywords:** sorghum, anthracnose, epicuticular wax, transcriptome analysis

## Abstract

Cuticular waxes are mixtures of hydrophobic compounds covering land plant surfaces and play key roles in plant resistance to abiotic and biotic stresses. However, it is still not clear whether the epicuticular wax could protect the plants from infection by anthracnose, one of the most important plant diseases worldwide, which seriously infects sorghum and causes great yield loss. In this study, *Sorghum bicolor* L., an important C4 crop with high wax coverage, was selected to analyze the relationship between epicuticular wax (EW) and anthracnose resistance. In vitro analysis indicated that the sorghum leaf wax significantly inhibited the anthracnose mycelium growth of anthracnose on potato dextrose agar (PDA) medium, with the plaque diameter smaller than that grown on medium without wax. Then, the EWs were removed from the intact leaf with gum acacia, followed by the inoculation of *Colletotrichum sublineola.* The results indicated that the disease lesion was remarkably aggravated on leaves without EW, which showed decreased net photosynthetic rate and increased intercellular CO_2_ concentrations and malonaldehyde content three days after inoculation. Transcriptome analysis further indicated that 1546 and 2843 differentially expressed genes (DEGs) were regulated by *C. sublineola* infection in plants with and without EW, respectively. Among the DEG encoded proteins and enriched pathways regulated by anthracnose infection, the cascade of the mitogen-activated protein kinases (MAPK) signaling pathway, ABC transporters, sulfur metabolism, benzoxazinoid biosynthesis, and photosynthesis were mainly regulated in plants without EW. Overall, the EW increases plant resistance to *C. sublineola* by affecting physiological and transcriptome responses through sorghum epicuticular wax, improving our understanding of its roles in defending plants from fungi and ultimately benefiting sorghum resistance breeding.

## 1. Introduction

Plants have evolved and acquired a sophisticated system to prevent plants from acquiring damage due to harsh environments and pathogen invasion [1]. The cuticular wax, as the first physical barrier of plant aerial parts, can protect plants against abiotic stresses such as drought and excessive ultraviolet radiation [2,3,4] and biotic stresses such as bacterial and fungal pathogen invasions [1,5]. However, some studies have shown that the cuticular waxes act as signaling molecules to favor fungal growth and development, resulting in appressorium formation and disease [5,6]. Anthracnose, one of the most important plant diseases worldwide, seriously infects plants and causes great yield loss. However, how the plant leaf cuticular wax influences anthracnose is still not clear.

Cuticular waxes are mixtures of hydrophobic compounds, mainly consisting of fatty acids and their derivatives such as alkanes, aldehydes, primary alcohols, alky esters, secondary alcohols, and ketones [7]. The composition and morphology of the cuticular waxes are very likely to influence leaf surface properties, which provide obstacles for invading fungi [5,8]. However, due to their differences in chain lengths, chemical structures, and polarities between compounds and plant species, their roles are inconsistent among plant species. For example, the very-long-chain aldehydes have been shown to induce appressorium formation in ascospores of the wheat powdery mildew fungus *Blumeria graminis* [9]. Therefore, the influence of cuticular waxes on certain pathogen fungi might be related to their chemical profiles.

*Colletotrichum sublineola* is an intracellular hemi-biotrophic fungal pathogen, a filamentous fungus that causes anthracnose disease in sorghum [10]. After invasion, the initial penetration event in sorghum begins with the formation of disease lesions and produces acervuli, which can germinate to produce germ tubes with appressoria, followed by the penetration of host epidermal cells [11]. Both environmental conditions and plant traits would influence the anthracnose. Warm and humid conditions can exacerbate anthracnose severity in sorghum [12]. The genetic diversity and agronomic traits would affect the sorghum resistance to anthracnose, which is a crucial genetic resource for breeding [13]. Chemicals, including 3′-deoxyanthocyanidins, anthocyanins, and flavonols, were identified to accumulate in vesicles and defend plants from fungal infection [10]. However, the functions of cuticular wax on anthracnose resistance is less explored.

Sorghum (*Sorghum bicolor* (L.) Moench) is an important C_4_ crop serving as a staple food and used in liquor brewing, in animal feed, or as bioenergy [14]. It is also one of the crops with higher wax coverages, mainly consisted of aldehydes and primary alcohols on the leaf and fatty acids on the sheath [5]. A study postulated that the epicuticular wax might act as the first physical barrier in defending the plant from *C. sublineola* infection and transition [8]. However, sorghum is one of the crops susceptible to *C. sublineola.* It was reported that anthracnose caused yield losses up to 67% in susceptible sorghum cultivars [15]. These results suggested that the high wax coverage on sorghum did not guarantee the sorghum disease resistance. Therefore, it is important to analyze the infection process of anthracnose and explore how sorghum plants have evolved and acquired defense mechanisms to protect themselves against fungal attacks. A deep understanding of the sorghum–anthracnose interactions is needed in order to rapidly identify resistance sources in the diverse landraces, isolation, and characterization of the R-genes [16].

The present study is aimed at exploring the importance of epicuticular wax in the infection processes of anthracnose *C. sublineola* in sorghum. The physiological and transcriptional responses of sorghum with or without epicuticular wax (EW) to anthracnose infection were analyzed in vitro and in vivo. This knowledge will resolve whether there is a correlation between the epicuticular wax and anthracnose susceptibility, forming the basis of comprehensive studies of defense responses and ultimately benefit sorghum resistance breeding.

## 2. Results

### 2.1. The Growth of C. sublineola on PDA Medium Amended with Wax

The *C. sublineola* strain was isolated from infected sorghum leaves in the field from Chongqing, China. The isolation was sequenced using primers ITS1/4 and ascertained as the causal agent of leaf blight by inoculating it on healthy sorghum leaves (Appendix A). Indeed, necrotic lesions appeared on sorghum leaves at seven days post inoculation (dpi), and the lesion area accounted for 10.5% of the whole leaf (Appendix A).

To analyze whether wax is important in defending a plant from anthracnose, the mycelium growth rate of *C. sublineola* was measured on PDA medium amended with wax in vitro (Figure 1A,B). Indeed, wax significantly inhibited the anthracnose mycelium growth, with the plaque diameter amended with wax significantly smaller (*p* < 0.05).

### 2.2. Physiological Responses of Sorghum to Anthracnose

Next, the EW of sorghum leaves was removed and incubated with anthracnose to measure its role in defending anthracnose in vivo. Actually, removing EW remarkably aggravated anthracnose infection, with clear and serious disease symptoms on leaves at 3 dpi (Figure 1C). The lesions were round or oval in shape, and the lesion area increased with brown in the center and orange-red on the edge. Anthracnose inoculation significantly decreased the net photosynthetic rate (Pn) and increased the intercellular CO_2_ concentrations (Ci) after removing EW at 3 dpi, while the Pn and Ci changed insignificantly for plant with EW (Figure 2A,B). After inoculation, the stomatal conductance (gs) and transpiration rate (Tr) decreased 35.73% and 39.99% in the plant with EW, and 10.79% and 9.97% in plants without EW, respectively (Figure 2C,D).

To unveil potential mechanisms of anthracnose-induced damage, the malonaldehyde (MDA) content was measured and it was significantly accumulated after anthracnose inoculation at 3 dpi for plant with or without EW (*p* < 0.05) (Figure 3A). Meanwhile, the activities of superoxide dismutase (SOD), catalase (CAT), and peroxidase (POD) were elevated with anthracnose inoculation at 3 dpi (*p* < 0.05) (Figure 4B–D). The inoculation of anthracnose in the −EW group increased the SOD, CAT, and POD activities by 1.34-, 1.66-, and 9.46-fold, respectively, and the change folds were 1.95, 15.21, and 5.68 for the +EW group, respectively. At 7 dpi, the MDA content was only accumulated in plants with EW, but the MDA content in the leaves without EW was higher than that in the leaves without EW under the condition of being sprayed with water (*p* < 0.05). The activities of CAT and SOD were still elevated with anthracnose inoculation at 7 dpi compared to those inoculated with water. The POD activity was only improved for the plants without EW at 7 dpi (*p* < 0.05).

### 2.3. Wax Content Changes after Anthracnose Infection

The total wax amount for plants with EW at 3 dpi was dramatically reduced by 35.14%, while there were no significant changes for the wax content in plants without EW. After anthracnose infection, total wax amounts in the sorghum leaves with EW reduced from 0.75 to 0.49 μg/cm^2^ (Figure 4A). However, the total wax content was slightly changed for plants without EW, with wax amounts ranging from 0.40 to 0.44 μg/cm^2^. Interestingly, the relative abundance of leaf wax classes changed in plants without EW (Figure 4 and Appendix A). The contents of alkanoic acids were 8.1% and 11.2% in leaves without EW treated with water and anthracnose, respectively. Other wax classes such as aldehydes (35.6% and 23.6%), primary alcohols (11.9% and 7.7%), alkanes (24.0% and 25.9%), and triterpenoids (3.1% and 4.6%) were also changed (Figure 4B). Overall, aldehyde content was dramatically reduced after anthracnose infection for plants with or without EW (Figure 4C).

### 2.4. Gene Expression Responses of Sorghum to Anthracnose

To analyze the differences in transcriptome anthracnose induced in sorghum with and without EW, RNA was isolated from sorghum leaves at 3 dpi, with no anthracnose inoculation as the control. The differentially expressed genes (DEGs) after anthracnose inoculation (FDR < 0.05 and *p* < 0.05) were defined. In total, 2843 genes in sorghum without EW showed altered expressions in response to anthracnose, including 1645 upregulated and 1198 downregulated DEGs (Figure 5A, Appendix A). The number of upregulated genes (861) in plants with EW was less than that in plants without EW (Figure 5B, Appendix A). Similarly, sorghum with EW displayed only 685 downregulated genes. Comparing the DEGs in these two groups, the number of specially identified upregulated and downregulated DEGs was 1187 and 809 in sorghum without EW, respectively (Appendix A). These results likely indicate that fungal growth in sorghum without EW induces and activates massive stress response genes, which accelerate the anthracnose infection process when compared with that in the sorghum with EW.

### 2.5. KEGG Enrichment Analysis of Differential Expression Pathways

KEGG pathway enrichment analysis was used to examine the possible biological function of genes identified as having differential expression in plants without EW compared to plants with EW (Figure 5C,D). The significantly enriched pathways regulated upon infection in the two groups included plant–pathogen interaction, plant hormone signal transduction, phenylpropanoid, biosynthesis of amino acid, flavonoid biosynthesis, circadian rhythm, and flavone and flavanol biosynthesis. The plant–pathogen interaction pathway plays important roles in defending from fungal infection and DEGs involved in the pathway showed a similar changing trend (Figure 6, Appendix A). Mapping the DEGs to the pathways also revealed that some of the genes only differentially expressed in plants without EW at 3 dpi, such as the cascade of the mitogen-activated protein kinases (MAPK) signaling pathway, ABC transporters, sulfur metabolism, benzoxazinoid biosynthesis, photosynthesis, etc. (Figure 5). Specifically, it was noticed that part of the genes in the MAPK signaling pathway only regulated in the plant without EW at 3 dpi (Figure 7).

### 2.6. Fatty-Acid-Synthesis-related Genes were Regulated

Pathways related to fatty acid elongation, plant–pathogen interactions, and cutin, suberin, and wax biosynthesis were observed to be differentially expressed (Table 1). Interestingly, one *fatty acyl-ACP thioesterases* (*FATB*) gene (*Sobic.010G180400*) encoding the palmitoyl-acyl carrier protein thioesterase showed elevated expression levels, and the expression of another *FATB* gene (*Sobic.010G033300*) was only induced in plants without EW (Table 1). Other fatty-acid-synthesis-related genes also showed differential expression, such as 3-ketoacyl-CoA synthase (KCS), very-long-chain 3-oxoacyl-CoA reductase (KCR), cytochrome P450 proteins (CYP), and so on (Table 1, Appendix A). To validate the RNA-seq results, four fatty-acid-related DEGs were randomly selected using quantitative real-time PCR (qRT-PCR). There is consistency between the RNA-seq results and the qRT-PCR results (Appendix A).

### 2.7. Correlation Analysis between Wax Content and Disease Index

To further validate the relationship between wax coverage and sorghum anthracnose, 30 sorghum cultivars were grown in fields where anthracnose frequently happens. The anthracnose infection index was negatively correlated with the amounts of total cuticular wax (R = −0.643, *p* < 0.05) (Figure 8, Appendix A). The anthracnose infection index was negatively correlated with the amounts of the contents of alkanes, aldehydes, and alkanoic acids (*p* < 0.05), but the absolute values of the R coefficient were lower than the R coefficient between the anthracnose infection index and the amounts of total cuticular wax.

## 3. Discussion

Plants encounter various pathogens, such as bacteria, viruses, and fungi, which have caused crop yield loss globally [17]. Anthracnose disease in sorghum, generally caused by *C. sublineola*, brings great damage to the production of sorghum, especially for susceptible sorghum cultivars [10,15]. In this study, the role of sorghum EW in defending plants from anthracnose was determined in vitro and in vivo. The mycelium growth rate of *C. sublineola* was reduced on PDA medium amended with wax in vitro. The EW slowed down the process of anthracnose infection in sorghum. These results suggested that the coating wax might act as a shield for anthracnose attack, protecting the plant from infection. During the infection processes, fungal pathogens could synthesize and secrete hydrolytic enzymes such as cutinases and lipases to degrade the cuticular wax layer [18]. For sorghum, the epicuticular wax might inhibit the growth of anthracnose, and thus its function of secreting hydrolytic enzymes.

Plants have evolved sophisticated mechanisms to protect themselves from biotic stresses [19]. Here, the study indicated that EW can prevent anthracnose growth, protect plants from fungal infection, and alleviate the fungal infection symptoms. Leaves without EW showed clear and serious disease symptoms and decreased net photosynthetic rate after 3 dpi of anthracnose inoculation. The symptoms were not obvious in the leaves with EW after 3 dpi of anthracnose inoculation, and the net photosynthetic rate changed insignificantly. Although the symptoms in plants with EW were slowed down, the MDA content was significantly accumulated and the activities of SOD, POD, and CAT all increased for both groups (+EW and −EW group) after anthracnose infection at 3 dpi. The enzymatic antioxidant system is linked to biotic stress tolerance in plants [20]. Enzyme activities of SOD, CAT, and POD are known to be induced by various pathogen infections, which would act as a form to defend the access of the pathogen to the membrane [21,22]. Although the symptom was not obvious in the group with EW, the enzymatic antioxidant system was triggered to defend the plant from fungal infection. The POD activity is significantly higher in leaves without EW (−EW_T) than that in leaves with EW (+EW_T) after inoculation with anthracnose at 3 dpi and 7 dpi. The induced level of POD enzymatic activities indicated that removing EW can partly accelerate anthracnose-induced oxidative stress.

Plant defense response was initiated from recognition of the pathogen and signaling transduction, followed by differential expression of genes and accumulation with metabolites [23]. A previous study revealed that the resistant sorghum had higher levels of H_2_O_2_ accumulation than the susceptible sorghum at 1 dpi and 3 dpi [8]. However, in our study more genes were regulated in the −EW group at 3 dpi. The infection was more severe in −EW sorghum, with larger lesion points and significantly reduced net photosynthetic rate. In that way, more regulated genes reflected the plant suffering a more severe fungal invasion in the −EW group than in the +EW group. The accelerated responses to anthracnose infection were supported by GO term annotation and KEGG pathway analysis, where plant defense processes were significantly enriched in plants without EW after anthracnose infection. The process of host receptors recognizes that the pathogen-associated molecular patterns (PAMPs) are involved in a wide range of defenses, including the production of ROS and the MAPKs [1,24]. The plant MAPK pathway plays important roles in defending plants against the pathogen attack processes [24]. In this study, the MAPK signaling pathway was only significantly accumulated in plants without EW after anthracnose infection (*p* < 0.05). A greater number of fungi-regulated genes in plants without EW in the MAPK signaling pathway probably indicated that removing EW would accelerate the anthracnose infection in sorghum. Due to the complexity of the defense response, several defense-related genes are involved in defense against pathogens. The defense-related genes are also involved in pathways of phenylpropanoid biosynthesis, ABC transporters, sulfur metabolism, and flavone and flavanol biosynthesis, which play important roles in anthracnose resistance.

The increase in alkanes has been shown to be a common response to drought stress [25,26,27,28]. However, the role of EW in defending plant from fungi is complex, depending on the plant and fungi species. It has been reported that sorghum leaf wax improved the growth of *Penicillium* but suppressed *Alternaria alternate*, whereas sheath wax suppressed *Penicillium* but did not affect *A. alternata* [5]. With the mutation of *MYB96* in *Medicago truncatula irg1/palm1* mutants, the composition of waxes on the abaxial leaf surface had >90% reduction in C_30_ primary alcohols, resulting in reduced spore differentiation of anthracnose and nonhost rust pathogens [29]. Here, the sorghum was more susceptible to anthracnose after removing the EW, with accelerated physiological and transcriptional responses. In addition, a negative correlation was observed between the wax content and anthracnose infection index in sorghum. These results suggested that the EW on sorghum leaves inhibited the anthracnose growth, slowed down anthracnose infection, and thus improved plant disease resistance. Therefore, in the field, selecting sorghum lines with relatively higher wax coverage might benefit both drought resistance and disease resistance. However, as mixtures of hydrophobic compounds, it is still difficult to state which wax compound or wax class is directly involved in anthracnose resistance. Further studies are needed to elucidate the antifungal activities of wax compounds and genes, which would facilitate breeding antifungal sorghum resources.

## 4. Materials and Methods

### 4.1. Plant Materials and Growth Conditions in Chamber

Sorghum (*Sorghum bicolor* L.) cv. Hongyingzi was used to analyze whether the epicuticular wax was important in the process of guarding plants from fungal infection. Plants were grown in pots in a growth chamber with a 10 h light/14 h dark cycle, at 28 °C to 25 °C (day to night) at 80% relative humidity, and the light intensity was 600 μmol/m^−2^/s^−1^. The pots (35 cm upper diameter × 20 cm lower diameter × 24 cm height) were filled with 2.2 kg of a mixture of sterilized peat soil and neutral loess soil (1:1, *v*/*v*). In total, there were sixteen pots with two plants in each pot. Plants were watered to keep the soil water content at 70–80% field capacity.

### 4.2. Isolation and Identification of C. sublineola

The diseased leaves of sorghum infected by *C. sublineola* were collected from the field of Beibei Research Station of Southwest University (29°48′ N, 106°25′ E), Chongqing, China. Tissues from the junction between diseased and healthy leaves were separated, washed with 75% alcohol, rinsed three times with sterile distilled water, and cultured on PDA medium at 28 °C. The growing colonies were transferred and purified, and single spore isolation was obtained. The genomic DNA of the isolated *C. sublineola* strain was extracted with the CTAB method [30], and the fragment was amplified using the primer ITS1/4 (forward primer 5′ TCCGTAGGTGAACCTGCGG 3′/ reverse primer 5′ TCC TCCGCTTATTGATATGC 3′), sequenced, and blasted on the database of the National Center for Biotechnology Information (NCBI). The blast result showed 98.43% similarity with the ITS sequence (GenBank: AB439813.1) of the identified *C. sublineola* strain MAFF 511474. The isolate was ascertained as the causal agent of leaf lesions of *C. sublineola* by inoculating it on healthy sorghum leaves. Based on the morphological characteristics and ITS sequence analysis, the isolate was identified as *C. sublineola* and the ITS sequence was submitted to GenBank (accession number: OQ271332).

### 4.3. Wax Bioassay

To test the role of wax on the growth in vitro, wax was extracted with chloroform from sorghum leaves at the jointing stage planted in the field and then 3 mg wax was dissolved in 3 mL chloroform and poured into the PDA medium to test its effects on the growth of fungi. Agar plugs (6 mm) were removed with a sterilized corn borer from the leading edge of the mycelia of *C. sublineola*, placed in the middle of the Petri dish with PDA, and incubated at 28 °C. The colony diameter was measured after 3 days. The PDA medium containing an equal amount of chloroform was used as control. Each treatment was replicated three times.

### 4.4. Fungal Infection Assays of Sorghum Cultured in Chamber

In order to analyze the role of EW in the process of protecting fungal infection in vivo, the sorghum plants at the sixth-leaf stage were inoculated with or without EW. The leaf surface was supplied with 90% (*w*/*w*) Arabic latex using a small soft brush and the EW was peeled off after a dry and stable polymer film was formed on the surface. This process was repeated three times. SEM also showed that three times of peeling could obtain almost all epicuticular wax. The peeled Arabic latex with epicuticular wax was further extracted with chloroform and the wax was purified for GC analysis.

The isolated strain of *C. sublineola* was propagated, sterilized with distilled water, and diluted to a final concentration of 1 × 10^6^ CFU/mL. The inoculation was finished by spraying the spore’s solution evenly onto the leaves. Two groups of plant leaves with or without epicuticular wax were sprayed with spores of *C. sublineola*, and the other two groups of leaves with or without epicuticular wax were sprayed with sterilized water as the control. The treatment groups and control groups were covered with a plastic film at 25 °C and 95% relative humidity in the incubator for 48 h. Then, the plants were transferred into the chamber. Each treatment was replicated three times. The pathogenicity of *C. sublineola* was analyzed and the 2nd and 3rd leaves were harvested on the 3rd day after inoculation with the *C. sublineola* spore suspension to detect enzyme activity assays and isolate the sample RNA for transcriptome analysis.

### 4.5. Photosynthetic Rate in Leaves and Measurement of Enzyme Activity Assay

After spraying the spore solution of fungi, the photosynthetic rate of the 2nd expanded leaves was measured with an LI-6400 portable photosynthesis system on 3 dpi and 7 dpi. The photosynthetically active radiation was set to 800 µmol m^−2^ sec^−1^. The freeze-dried and powered samples were used to analyze the content of MDA and enzyme assay activities of CAT, POD, and SOD [31].

### 4.6. Wax Extraction and GC-MS Analysis

Leaves (third leaf from the top) were taken on day 3 after fungi infection. Then, the surface wax mixtures were extracted with chloroform as described by Xiao et al. (2020). Before wax extraction, photos of leaves were taken and subjected to pixel counting using the ImageJ software to determine surface areas. Then, the surface wax mixtures were extracted twice for 30 s with CHCl_3_. The two extracts from each sample were then combined and filtered through glass wool, and the solvent was evaporated under N_2_. *n*-Tetracosane (5 µg) was added to the fresh plant material before extraction. The GC analysis was carried out with a 9790Ⅱ gas chromatograph (Fu-Li, China). The GC column was a DM-5 capillary column (30 m × 0.32 mm × 0.25 µm). Nitrogen served as the carrier gas. The GC oven was held at 80 °C for 10 min, then heated at 5 °C/min to 260 °C, where the temperature remained for 10 min. The temperature was then heated at 2 °C/min to 290 °C, and further heated at 5 °C/min to 320 °C, where the temperature was held for 10 min. Compounds were detected with a GCMS-QP2010 Ultra Mass Spectrometric Detector (Shimadzu Corp., Kyoto, Japan) using an HP-5 MS capillary column (30 m × 0.32 mm × 0.25 µm) and He as the carrier gas. Compounds were identified by comparing their mass spectra with published data and authentic standards, as well as the retention indices from NIST libraries (Appendix A).

### 4.7. RNA Isolation and Transcriptome Analysis

Total RNA was extracted from leaves (the 2nd and 3rd leaves from the top) using TRlzol Reagent according to the manufacturer’s instructions (Life technologies, Carlsbad, CA, USA). RNA integrity and concentration were checked using an Agilent 2100 Bioanalyzer (Agilent Technologies, Santa Clara, CA, USA). For RNA-seq library synthesis, three biological replicates per treatment were sequenced with an Illumina instrument. Gene expression levels were normalized by calculating reads per kilo base of transcript per million fragments mapped (RPKM) using HISAT2 after mapping to the Sorghum bicolor v3.1.1 (https://phytozome.jgi.doe.gov/pz/portal.html#!bulk?org=Org_Sbicolor) (accessed on 23 December 2021). The absolute value of log2 (Fold Change) ≥ 1 (treatment/control) (*p* ≤ 0.05) was defined as differentially expressed gene transcripts (DEGs). DEGs were also employed for Gene Ontology (GO) and Kyoto Encyclopedia of Genes and Genomes (KEGG) analysis. The qPCR was analyzed to verify the accuracy of the transcriptome data. The housekeeping sorghum gene *β-ACTIN* (*Sobic.001G112600*) was used as an internal standard (Appendix A).

### 4.8. Disease Index Survey

A total of 30 cultivars grown in the field were investigated during their jointing stages when symptoms of anthracnose disease with brownish and orange-reddish lesions on the leaves appeared. The leaves were sampled and analyzed for cuticular wax. The disease index was investigated and calculated according to the disease grading list (Appendix A) [32].
(1)$$\mathrm{Disease}\;\mathrm{index}=\frac{\sum (\mathrm{sample}\;\mathrm{number}\;\mathrm{of}\;\mathrm{each}\;\mathrm{disease}\;\mathrm{grade}\;\times \;\mathrm{representative}\;\mathrm{value}\;\mathrm{of}\;\mathrm{disease}\;\mathrm{grade})}{\mathrm{total}\;\mathrm{sample}\;\mathrm{number}\times \mathrm{representative}\;\mathrm{value}\;\mathrm{of}\;\mathrm{the}\;\mathrm{highest}\;\mathrm{disease}\;\mathrm{grade}}\times 100$$

### 4.9. Data Analysis

All data were presented as means ± SE. One-way ANOVA was applied to analyze the effects of fungal infection on plant physiological parameters and total wax according to the least-significant difference tests at the *p* < 0.05 level. Significance was defined as a probability level of the Student’s t test at *p* < 0.05. The Benjamini and Hochberg False Discovery Rate multiple testing correction was applied to all ANOVA analyses. Pearson correlation analysis was applied, and statistical significance was evaluated at *p* < 0.05.

## 5. Conclusions

We analyzed the physiological and transcriptional changes associated with the inoculation of *C. sublineola* in sorghum leaves with or without EW. Leaves without EW showed accelerated disease symptoms when compared with plants with EW, with clear disease spot, decreased net photosynthetic rate, and increased POD activity. The differential gene expression analysis identified over 2000 specific genes involved in anthracnose resistance. Gene Ontology and pathway enrichment analysis identified genes involved in the production of secondary metabolites, which are part of a typical host defense reaction. Overall, the EW increases plant resistance to *C. sublineola* by slowing down physiological damage, which improves our understanding of EW’s roles in defending plants from fungi and ultimately benefiting sorghum resistance breeding.

## Figures and Tables

**Figure 1 ijms-24-03070-f001:**
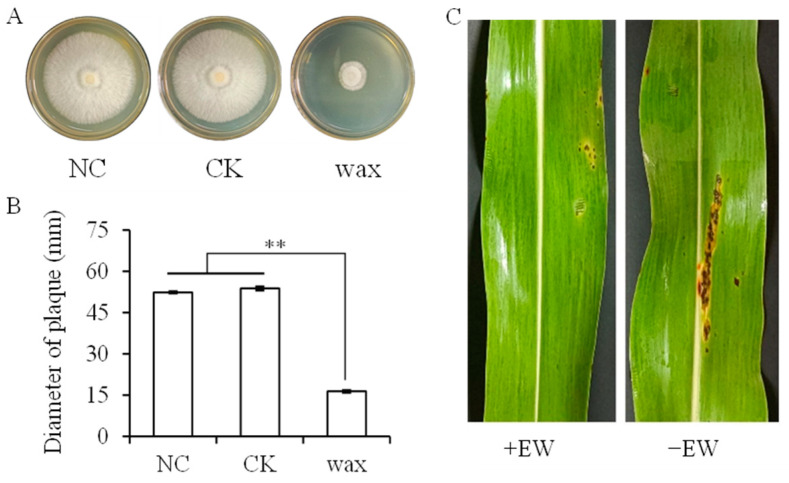
Diameters of anthracnose (*Colletotrichum sublineola*) colonies (**A**,**B**) on PDA medium amended with water (negative control, NC), chloroform (CK), or chloroform containing leaf wax (wax) and symptoms of sorghum leaves inoculating anthracnose remaining and removing epicuticular wax at 3 dpi (**C**). Bars with two stars (**) represent significance at *p* = 0.01 level according to least significant different test by a one-way analysis of variance with Duncan’s multiple range test (*n* = 6); +EW and −EW indicate remaining and removing epicuticular wax, respectively.

**Figure 2 ijms-24-03070-f002:**
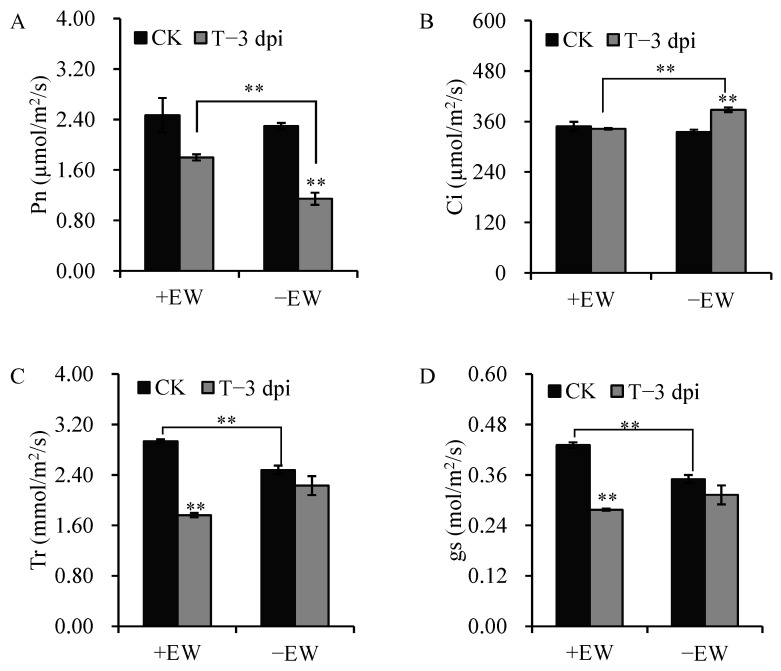
Effect of anthracnose on leaf net photosynthetic rate (Pn), intercellular CO_2_ concentrations (Ci), transpiration rate (Tr), and stomatal conductance (gs) as influenced by *Colletotrichum sublineola* infection in sorghum leaves at 3 dpi (T−3 dpi), with water as control (CK). (**A**) Effect of anthracnose on leaf Pn. (**B**) Effect of anthracnose on leaf Ci. (**C**) Effect of anthracnose on leaf Tr. (**D**) Effect of anthracnose on leaf gs. Bars with two stars (**) are significantly different at *p* ≤ 0.01 by a one-way analysis of variance with Duncan’s multiple range test, respectively. Values are means ± SE (*n* = 6).

**Figure 3 ijms-24-03070-f003:**
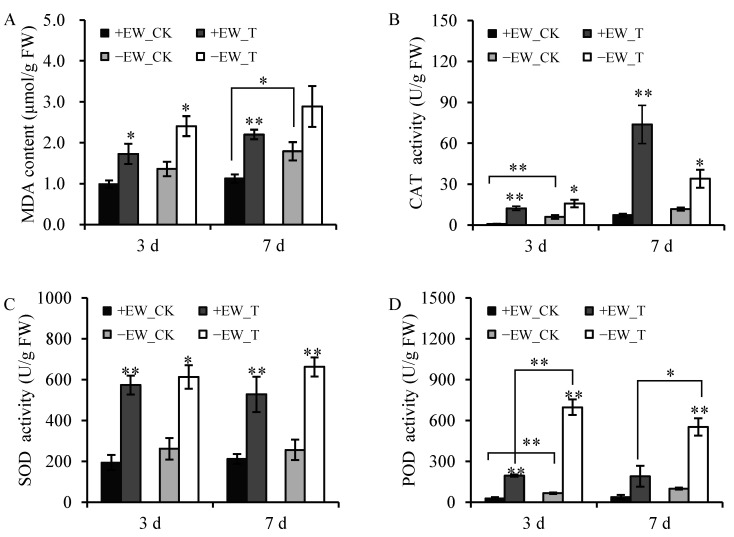
The MDA content (**A**), CAT activity (**B**), SOD activity (**C**), and POD activity (**D**) influenced by *Colletotrichum sublineola* infection in sorghum leaves, with water as control. Bars with one star (*) or two stars (**) are significantly different at *p* ≤ 0.05 or *p* ≤ 0.01, by a one-way analysis of variance with Duncan’s multiple range test, respectively. Values are means ± SE (*n* = 6).

**Figure 4 ijms-24-03070-f004:**
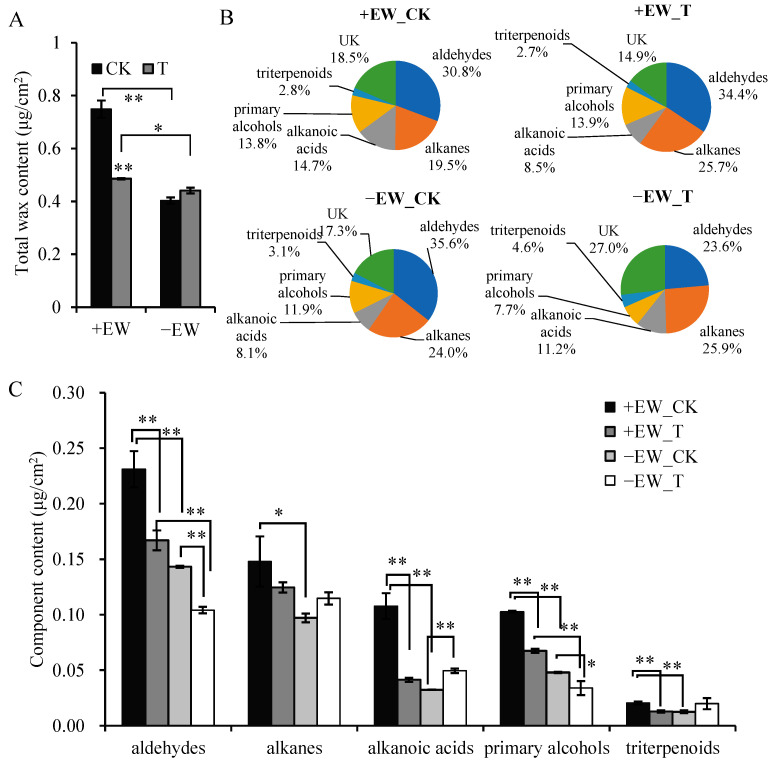
Responses of sorghum leaf wax monomer profiles to anthracnose infection. (**A**) Total wax amounts of leaves sampled from anthracnose infection of sorghum with or without epicuticular wax. (**B**) Pie plot of relative abundance of leaf wax monomer classes. (**C**) Relative abundance of individual compounds within wax monomer classes of alkanoic acids, aldehydes, primary alcohols, triterpenoids, and alkanes. Bars with one star (*) or two stars (**) are significantly different at *p* ≤ 0.05 or *p* ≤ 0.01, by a one-way analysis of variance with Duncan’s multiple range test, respectively. Values are means ± SE (*n* = 6).

**Figure 5 ijms-24-03070-f005:**
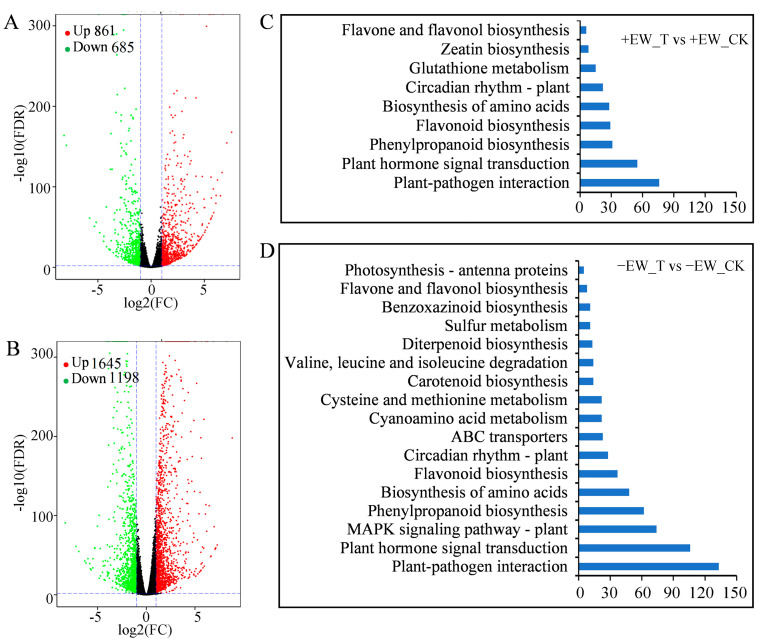
DEGs and their significantly enriched KEGG pathways of sorghum leaves to anthracnose infection. Volcano plot showing the DEGs from plants with epicuticular wax (**A**) and plants without epicuticular wax (**B**). Green represents downregulated genes and red represents upregulated genes. Significantly enriched KEGG pathways among the DEGs in plants with epicuticular wax (**C**) and removing epicuticular wax (**D**) (*q* < 0.05).

**Figure 6 ijms-24-03070-f006:**
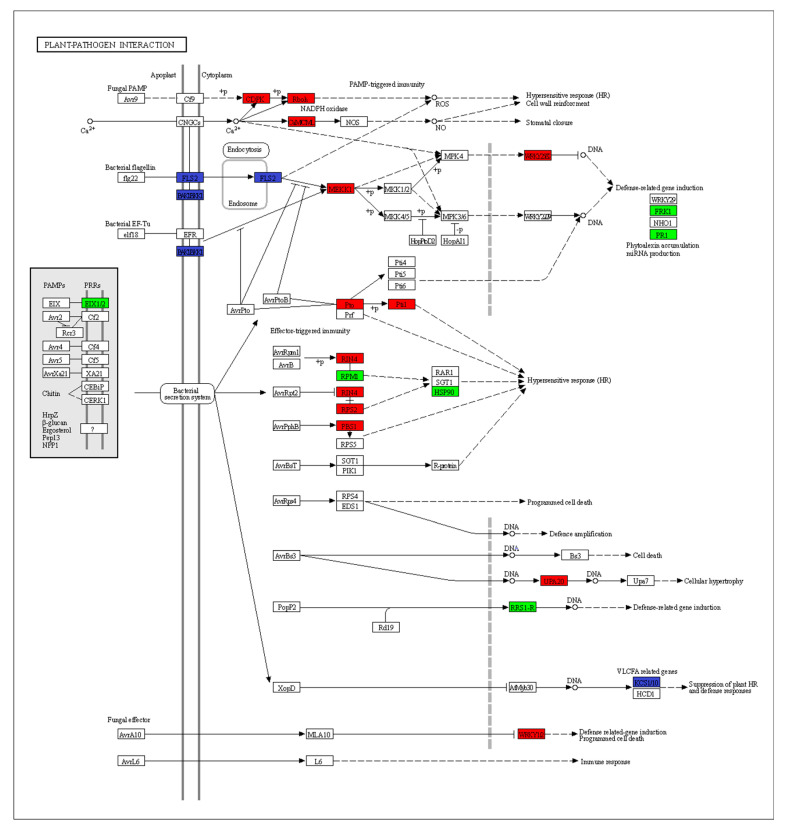
Differentially expressed genes are involved in plant–pathogen interactions. Expression values of the DEGs are presented as log_2_Fold Change > 1 (red for upregulated, green for downregulated, blue for both upregulated and downregulated).

**Figure 7 ijms-24-03070-f007:**
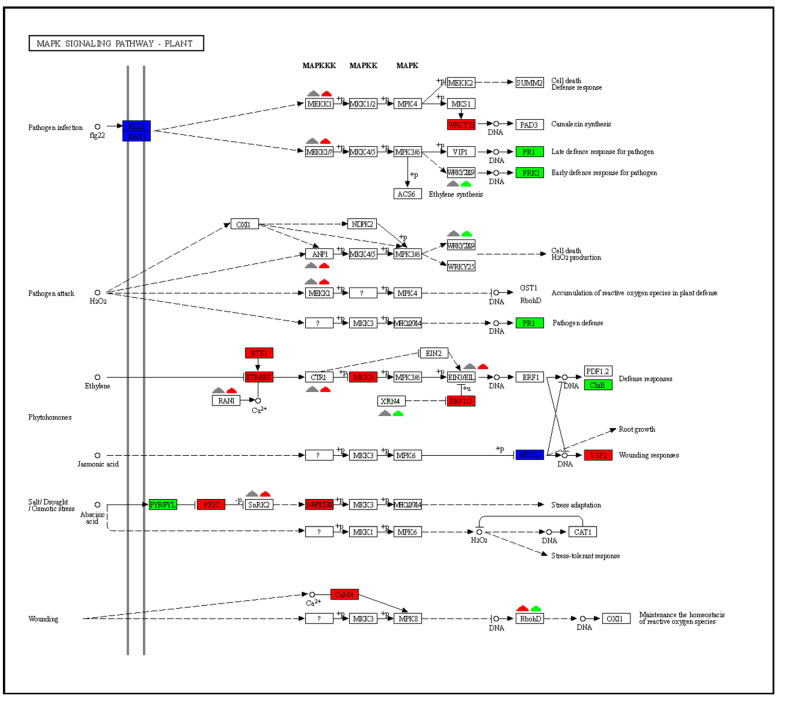
Differentially expressed genes are involved in MAPK signaling pathway. Expression values of the DEGs are presented as log_2_Fold Change > 1 (red for upregulated, green for downregulated, blue for both upregulated and downregulated, and gray for no significant changes). The rectangle represents common change trend, and the triangle represents different change trend. The first and second triangles represent plants with epicuticular wax and without epicuticular wax, respectively.

**Figure 8 ijms-24-03070-f008:**
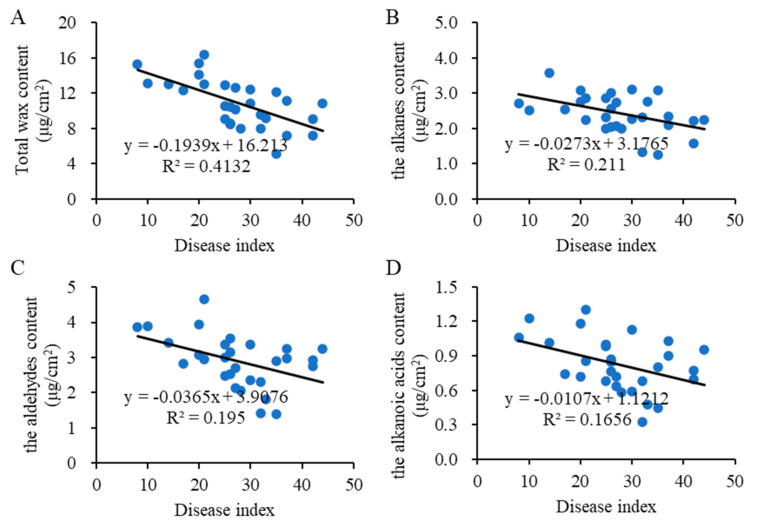
Correlation analysis between wax content and disease index. (**A**) Correlation analysis between total wax content and disease index. (**B**) Correlation analysis between alkanes content and disease index. (**C**) Correlation analysis between aldehydes content and disease index. (**D**) Correlation analysis between alkanoic acids content and disease index.

**Table 1 ijms-24-03070-t001:** Fatty-acid-synthesis-related genes regulated upon infection in sorghum at 3 days post inoculation: +EW indicates plants with epicuticular wax and −EW indicates the plants without EW. FC means Fold Change of gene expression level between treatment and its control. The red color represents Log_2_FC > 1, green color represents Log_2_FC < −1, and gray color represents −1 ≤ Log_2_FC ≤ 1.

Gene ID	Log2FC		Gene Annotation
	+EW	−EW	
*Sobic.010G180400*	1.1	1.7	Palmitoyl-acyl carrier protein thioesterase, FATB
*Sobic.010G033300*	0.2	2.2	Palmitoyl-acyl carrier protein thioesterase, FATB
*Sobic.001G438100*	1.2	1.0	3-ketoacyl-CoA synthase 1, KCS1
*Sobic.004G086800*	2.4	2.5	3-ketoacyl-CoA synthase 11, KCS11
*Sobic.004G249400*	−0.4	−1.4	3-ketoacyl-CoA synthase 5, KCS5
*Sobic.009G244300*	−0.2	−1.8	Protein RADIALIS-like 3, KCS17
*Sobic.004G267000*	−0.8	−2.5	Protein RADIALIS-like 3, KCS17
*Sobic.003G231800*	−0.3	−2.4	Protein RADIALIS-like 3, KCS17
*Sobic.008G114300*	−1.4	−1.3	Acetyl-CoA carboxylase 1, ACC1
*Sobic.006G125800*	1.1	−1.0	Very-long-chain 3-oxoacyl-CoA reductase 1, KCR1
*Sobic.002G207900*	1.9	−0.1	Very-long-chain aldehyde decarbonylase GL1-
*Sobic.004G064000*	1.2	0.1	Very-long-chain aldehyde decarbonylase GL1-2
*Sobic.001G231600*	−1.1	0.6	NADPH-dependent aldehyde reductase-like protein, ALDH
*Sobic.001G510300*	−1.8	0.1	Noroxomaritidine synthase 2, MAH1
*Sobic.001G451700*	0.6	1.8	Cytochrome P450 94A1, CYP94A1
*Sobic.006G186000*	1.1	1.2	Cytochrome P450 704C1, CYP704C1
*Sobic.010G055300*	−0.1	1.2	Cytochrome P450 94B1, CYP94B1
*Sobic.010G019500*	−0.5	−1.2	Cytochrome P450 704C1, CYP704C1

## Data Availability

This manuscript includes the essential data either as figures or as Appendix A.

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
