# Peer review of "The Effects of Epicuticular Wax on Anthracnose Resistance of Sorghum bicolor"

_ijms, 2023, doi:10.3390/ijms24043070_

Round 1
Reviewer 1 Report
In this study, the authors want to know the importance ofepicuticular wax during anthracnose infection processes.
The authors isolate and identify the pathogen of sorghum
anthracnose, observe the physiological responses, stress
related enzymes activities, wax content changes and the
differential expression genes. Based on those results,
the authors conclude that the EW increases plant resistance
to anthracnose by slowing down physiological damage, and suggest
the higher wax sorghum might provide disease resistance. Generally,
the work is of interest, the paper is well-written and meets the
scope of IJMS. However, some parts of manuscript still need to be
revised in more detail. Please check my questions and comments about
your manuscript below: Major comments: Line 86-90, 1. What's the BLAST result of ITS sequence, how to define the BLAST
result is correct or not? What's the current reliable methods to
identify Colletotrichum species? Please address those questions in
Results and Materials & methods sections. 2. Please submit the ITS sequence data to GenBank and provide
accession numbers. Line 305-309, 1. How to extract leaf wax, please briefly describe. 2. What's the purity of extracted leaf wax, is it only epicuticular
wax or containing other chemicals(like cuticle or cell wall or
others)? How to measure the purity of epicuticular wax ? 3. How to generate the -EW leaf, please briefly describe. 4. How to make sure the -EW leaf is 100% EW removed(means no EW)?
If the -EW leaf is only partial EW removed (ex. ~50%), is it
proper to use the word "without EW" or "-EW"? Line 213-218, 275-277 Only weak(R^2 < 0.5) or very weak(R^2 < 0.25) correlation between
wax content and disease index, authors should describe the
possibility more clearly and avoid jumping to conclusions without
solid evidence. Line 277-279, 376-378, The authors conclude that "The EW increases plant resistance to
C. sublineola by slowing down physiological damage". However,
more defense and stress related genes were up-regulated in the
-EW group at 3 dpi. In previous study (Basavaraju et al. 2009),
the resistant sorghum had higher level of H2O2 accumulation than
the susceptible sorghum at 1 dpi and 3 dpi. Which means higher level
of early responses can provide resistance(faster and stronger
responses in the resistant sorghum). Please discuss the difference
and possible reasons for the results. Minor comments: Fig. 1, 2, 3 and 4 1. Please briefly describe the statistical analysis method and
n value in figure legends. Fig. 4C Please label the statistical analysis results(one star (*) or two
stars (**)). Fig. 8, How to do the correlation analysis? How to define the high
correlation? Please describe in Materials & methods sections.
Author Response
Major comments:
Line 86-90, 1. What's the BLAST result of ITS sequence, how to define the BLAST result is correct or not? What's the current reliable methods to identify Colletotrichum species? Please address those questions in Results and Materials & methods sections.
Response: Good comments. The methods of Colletotrichum species identification were added in Results and Materials & methods sections.
We isolated the C. sublineola strain according to the methods described by Choi et al (2021). Tissues from the junction between disease and healthy leaves were separated and cultured on PDA medium at 28℃. The growing conidia showed sickle shape morphological characteristics were transferred and purified and single spore isolation was obtained. Cultures of the strain were maintained in liquid PDB on a rotary shaker at 160 rpm for 3 days. The genomic DNA of C. sublineola strain was extracted with the CTAB method, and the fragment was amplified using primer ITS1/4, sequenced, and blasted on the database of NCBI. The blast result showed 98.43% similarity with ITS sequence (GenBank: AB439813.1) of the identified C. sublineola strain MAFF 511474. The isolate was ascertained as the causal agent of leaf lesion of C. sublineola by inoculating it on healthy sorghum leaves. Based on the morphological characteristics and ITS sequence analysis, the isolate was identified as C. sublineola.
- Please submit the ITS sequence data to GenBank and provide accession numbers.
Response: The ITS sequence was submitted to GenBank as suggested. When we get the accession number, it will be added before publication.
Line 305-309, 1. How to extract leaf wax, please briefly describe.
Response: Detailed extraction method was added as suggested.
- What's the purity of extracted leaf wax, is it only epicuticular wax or containing other chemicals (like cuticle or cell wall or others)? How to measure the purity of epicuticular wax?
Response: More information related to epicuticular wax removal, extraction and purification were added in M&M. The wax extraction method is widely used all over the world, which could extract both epi- and intra-cuticular waxes. GC and GCMS analysis further indicated that cuticle and cell wall components were seldomly extracted.
- How to generate the -EW leaf, please briefly describe.
Response: The method to generate -EW leaf was provided in “4.4. Fungal infection assays of sorghum cultured in chamber”.
- How to make sure the -EW leaf is 100% EW removed (means no EW)? If the -EW leaf is only partial EW removed (ex. ~50%), is it proper to use the word "without EW" or "-EW"?
Response: The epicuticular wax was removed using Arabic latex. This process was repeated three times. SEM also showed that three times of peeling could obtain almost all epi-cuticular wax. The peeled Arabic latex with epicuticular wax was further extracted with chloroform and the wax was purified for GC analysis. See “4.4. Fungal infection assays of sorghum cultured in chamber”.
Line 213-218, 275-277 Only weak (R^2 < 0.5) or very weak (R^2 < 0.25) correlation between wax content and disease index, authors should describe the possibility more clearly and avoid jumping to conclusions without solid evidence.
Response:
Good comments. We revised the description as suggested.
Line 277-279, 376-378, The authors conclude that "The EW increases plant resistance to C. sublineola by slowing down physiological damage". However, more defense and stress related genes were up-regulated in the -EW group at 3 dpi. In previous study (Basavaraju et al. 2009), the resistant sorghum had higher level of H2O2 accumulation than the susceptible sorghum at 1 dpi and 3 dpi. Which means higher level of early responses can provide resistance (faster and stronger responses in the resistant sorghum). Please discuss the difference and possible reasons for the results.
Response: Added as suggested.
Previous study has revealed that the resistant sorghum had higher level of H2O2 accumulation than the susceptible sorghum at 1 dpi and 3 dpi [8]. However, in our study more genes were regulated in the -EW group at 3 dpi. The infection was more severe in -EW sorghum, with larger lesion points and significantly reduced net photosynthetic rate. In that way, more regulated genes reflected the plant suffering severer fungi invasion in the -EW group than in the +EW group.
Minor comments:
Fig. 1, 2, 3 and 4 1. Please briefly describe the statistical analysis method and n value in figure legends.
Response: Added as suggested.
Fig. 4C Please label the statistical analysis results (one star (*) or two stars (**)).
Response: Added as suggested.
Fig. 8, How to do the correlation analysis? How to define the high correlation? Please describe in Materials & methods sections.
Response: Pearson correlation analysis was applied and statistical significance was evaluated at P< 0.05. Added as suggested.
Reviewer 2 Report
The manuscript ijms-2152701, titled “The effects of epicuticular wax on anthracnose resistance of Sorghum bicolor”, addresses an important research paper showing the effect of epicuticular wax to overcome anthracnose. However, in my opinion, this paper must be revised in a major manner for reasons of forms and content. You will find below the comments.

Author Response
The manuscript ijms-2152701, titled “The effects of epicuticular wax on anthracnose resistance of Sorghum bicolor”, addresses an important research paper showing the effect of epicuticular wax to overcome anthracnose. However, in my opinion, this paper must be revised in a major manner for reasons of forms and content. You will find below the comments.
Line 20-23, Rephrase it to be one sentence.
Response: The sentences were rewritten.
Line 90, Can you add the lesion % (wax compared to negative control)
Response: Added as suggested.
Line 106, Can you specify symptoms (dieback, wilting, chlorosis or....) and how they progress.
Response: Added as suggested.
Line 111, It would better if you give % to see if there is any differences between treatment.
Response: Added as suggested.
Line 125, Can you add % (wax compared to negative control) of each enzyme, respectively.
Response: The changes of enzyme activities were presented with fold, rather than %. Added.
Line 148, how much did EW reduce the total wax.
Response: Reduced by 35.14%. Added.
Line 281, It would be better if you had measured PAL and phenol content.
Response: Good comments. It is commonly accepted that genes in the pathways of phenylpropanoid were regulated by disease. Here, we mainly focused on the physiological changes induced by removal of epicuticular wax. In our future studies, we will do further study to analyze the relationship between phenol and fungi infection.
Line 306, I suggest to add biomass result to see how much the use EW improves plant growth.
Response: Good comments. The biomass can reflect the effects of EW on anthracnose resistance of plant growth. In this study, the biomass did not change significantly between the groups with or without EW, probably due to relatively short experimental period. Indeed, we would like to analyze the biomass changes and long-term responses in our future study.
Line 310, Did sterilize this mixture or not?
Response: Yes, it was sterilized. Added.
Line 317, Did you rinse leaves after putting them in the alcohol with sterilize distilled water?
Response: Yes, we rinsed leaves with sterile distilled water after putting them in the alcohol. We have added it in the revised manuscript.
Line 392, Can you specify symptoms?
Response: Added as suggested.
Round 2
Reviewer 2 Report
Authors had taken all comments into consideration. However, i accept the publication of the present work.